

# Mortality in tongue cancer patients treated by curative surgery: a retrospective cohort study from CGRD

Ming-Shao Tsai[1,2], Chia-Hsuan Lai[3], Chuan-Pin Lee[2], Yao-Hsu Yang[2,4,5,6], Pau-Chung Chen[5,7], Chung-Jan Kang[8], Geng-He Chang[1], Yao-Te Tsai[1], Chang-Hsien Lu[9], Chih-Yen Chien[10], Chi-Kuang Young[11], Ku-Hao Fang[8], Chin-Jui Liu[1], Re-Ming A. Yeh[1] and Wen-Cheng Chen[2,3,12]

[1] Department of Otolaryngology—Head and Neck Surgery, Chiayi Chang Gung Memorial Hospital, Chiayi, Taiwan
[2] Center of Excellence for Chang Gung Research Datalink, Chiayi Chang Gung Memorial Hospital, Chiayi, Taiwan
[3] Department of Radiation Oncology, Chiayi Chang Gung Memorial Hospital, Chiayi, Taiwan
[4] Department of Traditional Chinese Medicine, Chiayi Chang Gung Memorial Hospital, Chiayi, Taiwan
[5] Institute of Occupational Medicine and Industrial Hygiene, National Taiwan University College of Public Health, Taipei, Taiwan
[6] School of Traditional Chinese Medicine, College of Medicine, Chang Gung University, Taoyuan, Taiwan
[7] Department of Environmental and Occupational Medicine, National Taiwan University Hospital, Taipei, Taiwan
[8] Department of Otolaryngology —Head and Neck Surgery, Linkou Chang Gung Memorial Hospital, Taoyuan, Taiwan
[9] Department of Medical Oncology, Chiayi Chang Gung Memorial Hospital, Chiayi, Taiwan
[10] Department of Otolaryngology, Kaohsiung Chang Gung Memorial Hospital, Kaohsiung, Taiwan
[11] Department of Otolaryngology —Head and Neck Surgery, Keelung Chang Gung Memorial Hospital, Keelung, Taiwan
[12] College of Medicine, Chang Gung University, Taoyuan, Taiwan

Corresponding author
Wen-Cheng Chen,
danielchen@cgmh.org.tw

## ABSTRACT

**Background**. Our study aimed to compare the outcomes of surgical treatment of tongue cancer patients in three different age groups.

**Methods**. From 2004 to 2013, we retrospectively analyzed the clinical data of 1,712 patients who were treated in the four institutions constituting the Chang Gung Memorial Hospitals (CGMH). We divided and studied the patients in three age groups: Group 1, younger (<65 years); Group 2, young old (65 to <75); and Group 3, older old patients (≥75 years).

**Results**. Multivariate analyses determined the unfavorable, independent prognostic factors of overall survival to be male sex, older age, advanced stage, advanced T, N classifications, and surgery plus chemotherapy. No significant differences were found in adjusted hazard ratios (HR) of death in early-stage disease (stage I–II) among Group 1 (HR 1.0), Group 2 (HR 1.43, 95% confidence interval (CI) [0.87–2.34], $p = 0.158$), and Group 3 (HR 1.22, 95% CI [0.49–3.03], $p = 0.664$) patients. However, amongst advanced-stage patients (stage (III–IV)), Group 3 (HR 2.53, 95% CI [1.46–4.38], $p = 0.001$) showed significantly worse survival than the other two groups after other variables were adjusted for. Fourteen out of 21 older old, advanced-staged patients finally died, and most of the mortalities were non-cancerogenic (9/14, 64.3%), and mostly occurred within one year (12/14, 85%) after cancer diagnosis. These non-cancer

cause of death included underlying diseases in combination with infection, pneumonia, poor nutrition status, and trauma.

**Conclusions**. Our study showed that advanced T classification (T3–4), positive nodal metastasis (N1–3) and poorly differentiated tumor predicted poor survival for all patients. Outcome of early-stage patients (stage I–II) among three age groups were not significantly different. However, for advanced-stage patients (stage III–IV), the older old patients ($\geq$75) had significantly worse survival than the other two patient groups. Therefore, for early-stage patients, age should not deny them to receive optimal treatments. However, older old patients ($\geq$75) with advanced cancer should be comprehensively assessed by geriatric tools before surgical treatment and combined with intensive postoperative care to improve outcome, especially the unfavorable non-cancerogenic mortalities within one year after cancer diagnosis.

# INTRODUCTION

The tongue is the most common site of intraoral cancers in Taiwan and most other countries (*Huang et al., 2008*; *Taiwan Ministry of Health and Welfare, 2016*; *Moore et al., 2000*). The tongue cancer patients are predominantly male, and incidence of tongue cancer peaks at 45–65 years in male and 55–75 years of age in female (*Taiwan Ministry of Health and Welfare, 2016*). According to a recent national cancer registry's annual report of Taiwan, incidence and mortality of head and neck cancers (ICD-O-3, C00-C14) rank sixth and fifth, respectively (*Taiwan Ministry of Health and Welfare, 2016*).

Taiwan, like the other developed countries, has gradually transformed into a society of the aged with those older than 65 years accounting for 12% of its population (*Kowal et al., 2012*; *Taiwan National Development Council, 2014*). This figure is likely to hit 20% in 2025, turning Taiwan into a super-ageing society (*Taiwan National Development Council, 2014*). The numbers of elderly patients with tongue cancer is expected to increase in the future. Nowadays, surgery is the therapeutic mainstay for early-stage tongue cancer, but it is often part of a multi-modal approach to treat advanced disease (*Calabrese et al., 2011*). However, many elderly patients may not be considered as candidates for aggressive multimodal treatments due to other ageing-associated comorbidities, general debility, and concerns regarding low tolerance to treatment and resulting toxicity (*Siddiqui & Gwede, 2012*; *Zabrodsky et al., 2004*).

Recent reports on the relationship between elderly patients with head and neck cancer and their prognosis have been conflicting (*Airoldi et al., 2004*; *Bhattacharyya, 2003*; *Chang et al., 2013*; *Clayman et al., 1998*; *Italiano et al., 2008*; *Kruse et al., 2010*; *Luciani et al., 2010*; *Lusinchi et al., 1990*; *Ortholan et al., 2009*; *Sarini et al., 2001*; *Zabrodsky et al., 2004*). Some concluded that older patients suffered a worse survival than younger patients (*Bhattacharyya, 2003*; *Chang et al., 2013*; *Clayman et al., 1998*). However, many others failed to show a significant difference between outcomes of old and young patients (*Airoldi*

*et al., 2004*; *Argiris et al., 2004*; *Lusinchi et al., 1990*; *Sarini et al., 2001*). In many previously published reports (*Airoldi et al., 2004*; *Bhattacharyya, 2003*; *Chang et al., 2013*; *Clayman et al., 1998*; *Italiano et al., 2008*; *Kruse et al., 2010*; *Lusinchi et al., 1990*; *Ortholan et al., 2009*; *Sarini et al., 2001*; *Zabrodsky et al., 2004*), the cutoff age values (65, 70, 75, 80, or 85 years) and the definition of elderly patients were inconsistent. Besides, some previous reports included a small sample size or lacked cancer staging. In the present study, we intend to focus on the survival outcomes of older adults with homogenous tongue cancer receiving curative surgery in order to provide evidence for preoperative risk explanations and decision making for the surgeons or oncologists. The National Institute of Aging have classified the elderly patients into three age groups: 65–74 years as "young old," 75–84 years as "older old," and >85 years as "oldest old" (*NIH, 1998*).

Here, we compared treatment results of tongue cancer patients, stratifying by three age groups: Group 1, <65 years (younger population); Group 2, 65–<75 years (young old population) and Group 3, ≥75 years (older old population).

## MATERIALS AND METHODS

### Data source

The data were obtained from the largest private hospital system in Taiwan, the Chang Gung Memorial Hospital (CGMH), using the Chang Gung Research Database (CGRD). The database combines original medical record from four medical institutes, Keelung CGMH, Linkou CGMH, Chiayi CGMH, and Kaohsiung CGMH. They are located in the northeast, northern, central, and southern regions of Taiwan, respectively. According to the Taiwanese national cancer registry's report, this combined hospital system had treated ∼20% of head and neck cancer patients.

We retrospectively reviewed the CGRD database from January 2004 to December 2013, and retrieved data of tongue cancer patients ($n = 2,487$). We excluded patients with recurrent or secondary oral cancers, or those with other malignancies ($n = 471$). Patients with poor performance status (ECOG $\geq 3$), end-stage renal disease, Child-Pugh C liver cirrhosis, or poor heart or lung function, or who were unfit for surgery, were also excluded to reduce confounding factors and bias. Finally, 1,712 patients with primary tongue cancer who received curative surgery were studied. The ethics review board of our institution approved the study (CGMH-IRB No. 104-4642B).

### Surgery, adjuvant therapy, and follow-up

Patients were evaluated preoperatively according to the CGMH oral cavity cancer guidelines, which were modified from the NCCN guideline (*Pfister et al., 2000*; *Pfister et al., 2013*). Evaluations included patient history taking, physical examination, nasopharyngoscopy, complete blood count, blood biochemistry, chest X-ray, electrocardiography, abdominal sonography and panendoscopy, computed tomography or MRI of head and neck, and bone scan or FDG-PET. Cancer staging accorded with the American Joint Committee on Cancer staging classification (AJCC, 6th edition) (*Edge & Compton, 2010*).

All patients were treated based on the CGMH oral cavity cancer guidelines. Tumors were resected with at least 1 cm gross, safe margin in all patients. Level I–III cervical lymph node

resections were performed in patients without lymph node metastases. Level I–V cervical lymph node resections or more extensive resections were done in patients with lymph node metastases. All patients who received free-flap reconstruction were admitted to the ICU after surgery, and were followed by intensive flap monitoring. Post-operative concurrent chemo-radiotherapy (CCRT) with 60–70 Gray (1.8–2.0 Gray per fraction) and Cisplatin-based regimen (weekly 30–40 mg/m2/wk × 6–7 weeks) was administered in patients with positive surgical margins or extracapsular extension of lymph nodes. In patients with other risk factors (such as T3/T4, N1, N2/N3, perineural invasion, or vascular tumor embolism), postoperative radiotherapy (RT) with 60–70 Gray (1.8–2.0 Gray per fraction) was administered. However, in older patients or patients with multiple comorbidities, RT or chemotherapy (CT) programs were cancelled following discussions with their families. We followed up the patients since their cancers' diagnosis until death, cancer recurrence, or the last follow-up. All patients received regular postoperative follow-up.

### Age definitions, outcomes, and covariates
The final dataset was divided into three groups: Group 1, younger population (< 65 years); Group 2, young old population (65 to <75 years); and Group 3, older old population (≥75 years). Patient characteristics included age, gender, cancer TNM staging, histological grade, treatment modalities (surgery, surgery with adjuvant RT or CT). The main outcome was overall survival rate.

### Statistical analysis
Gender, cancer staging, histological grade, and treatment modalities were compared amongst the three Groups by the Pearson's $\chi^2$ test. We estimated the survival rates during the entire follow-up period by the Kaplan–Meier method and compared survival rates amongst the three groups by the Log-rank test. The multivariate Cox proportional hazards analysis was performed for gender, age group, cancer stage, histological grade and treatment modalities. Statistical analyses used the statistical software R (version 3.1.3). For all tests, significance was defined at $p < 0.05$.

## RESULTS
### Patient characteristics and treatments
Patients' characteristics are presented in Table 1. The mean/median ages for Group 1, Group 2, and Group 3 were 48.7/49, 68.7/69, and 79.5/79 years, respectively. The proportion of female patients increased with age and were significantly different in three groups ($p < 0.001$). Cancer staging, T–N classification, and tumor differentiation were not significantly different among the three Groups. The ratio of patients receiving surgery alone without CT or RT increased with age and were significantly different amongst the three groups ($p = 0.004$). To clarify the difference in treatment patterns among the three groups, all patients were further divided into early (stage I–II) and advanced stages (stage III–IV) for comparison (Table 2). For early-stage patients, 86.7%, 85%, and 81% received surgery alone in Group 1, Group 2, and Group 3, respectively. The treatment patterns were not significantly different among the three Groups ($p = 0.558$). However, in patients

**Table 1** Clinicopathological characteristics of 1,712 patients with oral tongue cancer receiving surgery stratified by three age groups.

| | Age < 65 $n = 1,476$ (86.2%) | | Age 65 to <75 $n = 178$ (10.4%) | | Age ≥ 75 $n = 58$ (3.4%) | | p-value |
|---|---|---|---|---|---|---|---|
| Mean age (SD) | 48.7 (8.5) | | 68.7 (2.7) | | 79.5 (3.9) | | |
| Median (range) | 49 (21–64) | | 69 (65–74) | | 79 (75–92) | | |
| **Gender** | | | | | | | **<0.001** |
| Male | 1,323 | 89.6% | 145 | 81.5% | 36 | 62.1% | |
| Female | 153 | 10.4% | 33 | 18.5% | 22 | 37.9% | |
| **Stage** | | | | | | | 0.829 |
| I | 521 | 35.3% | 67 | 37.6% | 23 | 39.7% | |
| II | 346 | 23.4% | 46 | 25.8% | 14 | 24.1% | |
| III | 202 | 13.7% | 22 | 12.4% | 9 | 15.5% | |
| IV | 407 | 27.6% | 43 | 24.2% | 12 | 20.7% | |
| **T classification** | | | | | | | 0.151 |
| 1 | 558 | 37.8% | 74 | 41.6% | 25 | 43.1% | |
| 2 | 534 | 36.2% | 70 | 39.3% | 25 | 43.1% | |
| 3 | 138 | 9.3% | 13 | 7.3% | 5 | 8.6% | |
| 4 | 246 | 16.7% | 21 | 11.8% | 3 | 5.2% | |
| **N classification** | | | | | | | 0.716 |
| 0 | 1,025 | 69.4% | 134 | 75.3% | 43 | 74.1% | |
| 1 | 160 | 10.8% | 14 | 7.9% | 6 | 10.3% | |
| 2 | 288 | 19.5% | 30 | 16.9% | 9 | 15.5% | |
| 3 | 3 | 0.2% | 0 | 0.0% | 0 | 0.0% | |
| **Histological grade** | | | | | | | 0.847 |
| Well | 444 | 30.1% | 55 | 30.9% | 17 | 29.3% | |
| Moderately | 896 | 60.7% | 103 | 57.9% | 34 | 58.6% | |
| Poorly | 136 | 9.2% | 20 | 11.2% | 7 | 12.1% | |
| **Treatment** | | | | | | | 0.004 |
| Surgery alone | 868 | 58.8% | 120 | 67.4% | 44 | 75.9% | |
| Surgery + CT or RT | 608 | 41.2% | 58 | 32.6% | 14 | 24.1% | |
| RT alone | 180 | 12.2% | 23 | 12.9% | 8 | 13.8% | |
| CT alone | 58 | 3.9% | 8 | 4.5% | 2 | 3.4% | |
| CCRT | 370 | 25.1% | 27 | 15.2% | 4 | 6.9% | |

Notes.
SD, Standard deviation; CT, Chemotherapy; RT, Radiotherapy; CCRT, Concurrent chemo-radiotherapy.

with advanced-stage disease, adjuvant treatment (which were usually needed for control of the advanced stage) was given to fewer patients with increasing age. The proportion of advanced stage patients receiving adjuvant treatment were significantly different among the three groups ($p < 0.001$).

## Survival

The medial follow-up times of all and surviving patients are 2.88 and 3.66 years, respectively. Figure 1A shows the early-stage patients' overall survival curves. Survival rates were not

**Table 2  Characteristics and treatments of early stage (Stage I–II) and advanced stage (Stage III–IV) patients stratified by three age groups.**

| | Stage I-II | | | P-value | Stage III-IV | | | P-value |
|---|---|---|---|---|---|---|---|---|
| | <65 | 65 to <75 | ≥75 | | <65 | 65 to <75 | ≥75 | |
| Numbers | 867 (85.3%) | 113 (11.1%) | 37 (3.6%) | | 609 (87.6%) | 65 (9.4%) | 21 (3.0%) | |
| Mean age (±SD) | 48.6 (±8.6) | 68.8 (±2.6) | 80.0 (±4.3) | | 48.8 (±8.3) | 68.6 (±2.9) | 78.6 (±2.8) | |
| Median (range) | 49 (21–64) | 69 (65–74) | 79 (75–92) | | 49 (27–64) | 69 (65–74) | 79 (75–85) | |
| Gender | | | | <0.001 | | | | <0.001 |
| Male | 763 (88.0%) | 92 (81.4%) | 21 (56.8%) | | 560 (92.0%) | 53 (81.5%) | 15 (71.4%) | |
| Female | 104 (12.0%) | 21 (18.6%) | 16 (43.2%) | | 49 (8.0%) | 12 (18.5%) | 6 (28.6%) | |
| Treatment | | | | 0.558 | | | | <0.001 |
| Surgery alone | 752 (86.7%) | 96 (85.0%) | 34 (91.9%) | | 116 (19.0%) | 24 (36.9%) | 10 (47.6%) | |
| Surgery + CT or RT | 115 (13.3%) | 17 (15.0%) | 3 (8.1%) | | 493 (81.0%) | 41 (63.1%) | 11 (52.4%) | |
| RT alone | 81 (9.3%) | 9 (8.0%) | 3 (8.1%) | | 99 (16.3%) | 14 (21.5%) | 5 (23.8%) | |
| CT alone | 15 (1.7%) | 3 (2.7%) | 0 (0.0%) | | 43 (7.1%) | 5 (7.7%) | 2 (9.5%) | |
| CCRT | 19 (2.2%) | 5 (4.4%) | 0 (0.0%) | | 351 (57.6%) | 22 (33.8%) | 4 (19.0%) | |

**Notes.**
SD, Standard deviation; CT, Chemotherapy; RT, Radiotherapy; CCRT, Concurrent chemo-radiotherapy.
The median follow-up time was 40.0 months (range 0.3–98.2 months).

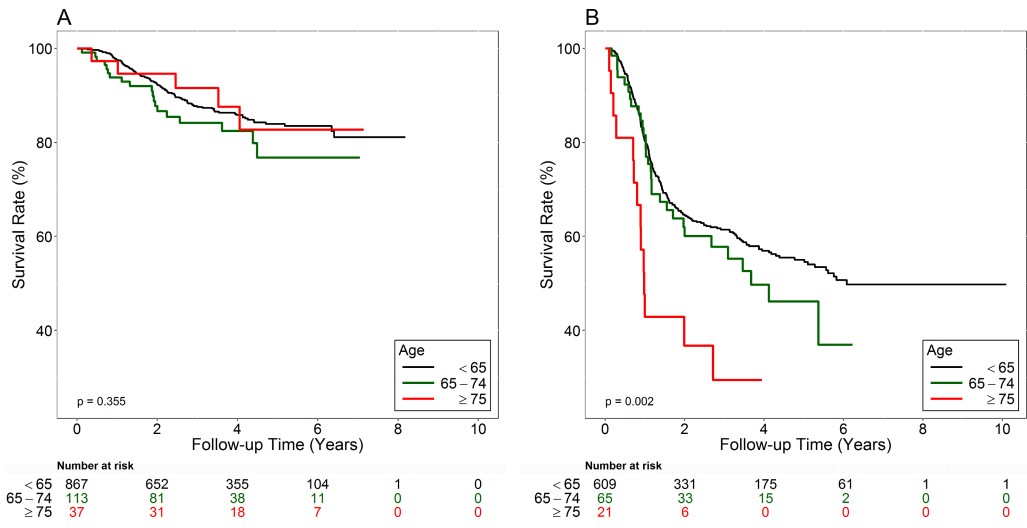

**Figure 1  Overall survival curves of early-stage (A) and advanced-stage (B) patients.**

significantly different amongst Group 1, Group 2, and Group 3 patients (Log-rank test, $p = 0.355$). The 5-year survival rates were 83.9% in Group 1; 76.8% in Group 2; and 82.7% in Group 3.

Figure 1B shows the advanced-stage patients' overall survival curves. Group 3 had the worst prognosis (Log-rank test, $p = 0.002$). The 5-year survival rate was 55.0% in Group 1, 46.1% in Group 2, and 29.4% in Group 3. Table 3 shows the multivariate analysis data used to compare the hazard ratios (HR) of death between different genders, the three Groups, stages (I and II *vs* III and IV), T classifications (T1–2 *vs* T3–4), N classifications (N0 *vs* N1–3), tumor differentiation (well to moderately differentiated *vs* poorly differentiated),

**Table 3** Multivariate analyses of risk factors regarding overall survival of all patients ($n = 1,712$) using Cox Proportional Hazard Model.

| Covariate | HR | CI(95%) | | P-value |
|---|---|---|---|---|
| Gender (ref: female) | | | | |
|     Male | 1.39 | 1.00 | 1.93 | **0.048** |
| Age (ref: <65) | | | | |
|     65–<75 | 1.38 | 1.02 | 1.87 | **0.037** |
|     $\geq 75$ | 1.92 | 1.20 | 3.06 | **0.007** |
| Stage (ref: I+II) | | | | |
|     III+IV | 1.64 | 1.07 | 2.50 | **0.022** |
| T classification (ref: $T_1$, $T_2$) | | | | |
|     $T_3$, $T_4$ | 1.42 | 1.10 | 1.84 | **0.008** |
| N classification (ref: No) | | | | |
|     $N_1$, $N_2$, $N_3$ | 2.41 | 1.70 | 3.42 | **<0.001** |
| Histological grade (ref: WD, MD) | | | | |
|     PD | 1.51 | 1.16 | 1.97 | **0.003** |
| Treatment (ref: surgery alone) | | | | |
|     Surgery + RT alone | 1.07 | 0.77 | 1.48 | 0.683 |
|     Surgery + CT alone | 1.97 | 1.33 | 2.90 | **0.001** |
|     Surgery + CCRT | 0.93 | 0.69 | 1.26 | 0.645 |

**Notes.**

HR, Hazard ratio; CI, Confidence interval; WD, Well differentiated; MD, Moderately differentiated; PD, Poorly differentiated; RT, Radiotherapy; CT, Chemotherapy; CCRT, Concurrent chemo-radiotherapy.

and treatments (Surgery *vs* Surgery + RT *vs* Surgery + CT *vs* Surgery + CCRT). For all patients, HR for death was 1.39 times greater for the male than for the female patients (95% CI [1.00–1.93]; $p = 0.048$). After adjusting for other factors, Group 2 (HR 1.38, 95% CI [1.02–1.87], $p = 0.037$) and Group 3 (HR 1.91, 95% CI [1.2–3.06], $p = 0.007$) had greater likelihood of death than Group 1. Advanced-stage (stage III–IV) patients had the worse survival (HR 1.64, 95% CI [1.07–2.50], $p = 0.022$) than patients with early-stage diseases (stage I–II). Adjusted HR for death were 1.42 times (95% CI [1.10–1.84], $p = 0.008$) and 2.41 times (95% CI [1.7–3.42], $p < 0.001$) for the advanced T (T3–4) and N (N1–3) classifications than for the early T (T1–2) and N0. Poorly differentiated tumors also predicted poor survival rates than well differentiated and moderately differentiated tumors (HR 1.51, 95% CI [1.16–1.97], $p = 0.003$). Surgery plus CT alone led to worse survival rates than surgery alone (HR 1.97, 95% CI [1.33–2.90], $p = 0.001$). Table 4 shows HR of death for early-stage (stage I and II) patients. The adjusted HR for death was 1.64 times greater for the male than for the female patients (95% CI [0.92–2.94], $p = 0.095$). The adjusted HR for death was greater for the Group 2 (HR 1.43, 95% CI [0.87–2.34], $p = 0.158$) and Group 3 (HR 1.22, 95% CI [0.49–3.03], $p = 0.664$) than for Group 1, but the differences were not statistically significant. Table 5 shows the HR of death for advanced-stage (stage III and IV) patients. The adjusted HR for death was 1.16 times greater for the male than for the female patients (95% CI [0.78–1.72], $p = 0.454$). The adjusted HR for death was significantly greater for Group 3 (HR 2.53, 95% CI [1.46–4.38], $p = 0.001$) than for Group 1, but the

**Table 4** Multivariate analyses of risk factors regarding overall survival of early-stage (I–II) patients ($n = 1,017$) using Cox Proportional Hazard Model.

| Covariate | HR | CI(95%) | | P-value |
|---|---|---|---|---|
| Gender (ref: female) | | | | |
| Male | 1.64 | 0.92 | 2.94 | 0.095 |
| Age (ref: <65) | | | | |
| 65–<75 | 1.43 | 0.87 | 2.34 | 0.158 |
| ≥ 75 | 1.22 | 0.49 | 3.03 | 0.664 |
| Histological grade (ref: WD, MD) | | | | |
| PD | 1.81 | 1.00 | 3.28 | 0.051 |
| Treatment (ref: surgery alone) | | | | |
| Surgery + RT alone | 1.95 | 1.19 | 3.22 | **0.009** |
| Surgery + CT alone | 3.63 | 1.68 | 7.86 | **0.001** |
| Surgery + CCRT | 4.04 | 2.03 | 8.01 | **<0.001** |

**Notes.**
HR, Hazard ratio; CI, Confidence interval; WD, Well differentiated; MD, Moderately differentiated; PD, Poorly differentiated; RT, Radiotherapy; CT, Chemotherapy; CCRT, Concurrent chemo-radiotherapy.

**Table 5** Multivariate analyses of risk factors regarding overall survival of advanced stage (III–IV) patients ($n = 695$) using Cox Proportional Hazard Model.

| Covariate | HR | CI(95%) | | P-value |
|---|---|---|---|---|
| Gender (ref: female) | | | | |
| Male | 1.16 | 0.78 | 1.72 | 0.454 |
| Age (ref: <65) | | | | |
| 65–<75 | 1.19 | 0.81 | 1.75 | 0.372 |
| ≥ 75 | 2.53 | 1.46 | 4.38 | **0.001** |
| Histological grade (ref: WD, MD) | | | | |
| PD | 1.46 | 1.08 | 1.97 | **0.013** |
| Treatment (ref: surgery alone) | | | | |
| Surgery + RT alone | 0.65 | 0.44 | 0.95 | **0.028** |
| Surgery + CT alone | 1.62 | 1.06 | 2.47 | **0.026** |
| Surgery + CCRT | 0.89 | 0.67 | 1.19 | 0.424 |

**Notes.**
HR, Hazard ratio; CI, Confidence interval; WD, Well differentiated; MD, Moderately differentiated; PD, Poorly differentiated; RT, Radiotherapy; CT, Chemotherapy; CCRT, Concurrent chemo-radiotherapy.

difference was not significant for Group 2 (HR 1.19, 95% CI [0.81–1.75], $p = 0.372$) as compared with Group 1.

## Causes of death in Group 3 patients with advanced disease

Fourteen of the 21 advanced-staged, Group 3 patients died. Causes of death are listed in Table 6. Nearly all mortalities (12 out of 14, 85.7%) occurred within 1 year after cancer diagnosis. The causes of death were classified into cancer recurrence (four patients, 28.6%), non-cancerogenic cause of death (nine patients, 64.3%), and unknown (one patient, 7.1%). The non-cancer cause of death are the primary causes of death in this age group, including underlying diseases in combination with infection, pneumonia, poor nutrition status, and trauma. Sixteen patients (16/21, 76.2%) were indicated to receive adjuvant therapy;

**Table 6  Causes of mortality of very old patients with stage III–IV tongue cancer ($n = 21$).**

| No | Sex | Age | Survival time (days) | Adjuvant (needed/done) | Cause of death | Details |
|----|-----|-----|----------------------|------------------------|----------------|---------|
| 1 | F | 79 | 38 | Y/N | Non-cancer | Sepsis, acute renal failure, pneumonia, malnutrition, type II DM (die on post-op day 10) |
| 2 | M | 76 | 53 | Y/N | Non-cancer | Pneumonia, respiratory failure |
| 3 | M | 81 | 76 | Y/N | Non-cancer | Severe hyponatremia caused by syndrome of inappropriate antidiuretic hormone secretion (SIADH) |
| 4 | F | 77 | 102 | Y/N | Non-cancer | Pneumonia, DM, HT |
| 5 | M | 85 | 257 | N/N | Non-cancer | Pneumonia, poor renal function, COPD, DM, HT, anemia, |
| 6 | M | 76 | 264 | Y/Y | Cancer | Multiple bone metastasis, poor intake, hospice |
| 7 | M | 78 | 294 | Y/Y | Unknown | Medical record of death in other hospital |
| 8 | M | 76 | 323 | Y/Y | Cancer | Neck local recurrence and pneumonia |
| 9 | M | 79 | 326 | Y/N | Cancer | Cancer recurrence, cachexia, COPD, DM, major depression |
| 10 | F | 79 | 329 | Y/Y | Cancer | Lung metastasis, neck metastasis and trachea invasion with bleeding |
| 11 | M | 81 | 354 | N/N | Non-cancer | Pneumonia, sepsis, type II DM, renal failure |
| 12 | M | 82 | 365 | N/N | Non-cancer | Atrial fibrillation and flutter |
| 13 | F | 79 | 727 | Y/Y | Non-cancer | Pneumonia |
| 14 | M | 75 | 992 | Y/Y | Non-cancer | Fall down and femoral fracture, sepsis, poor nutrition, hypokalemia |
| 15 | F | 82 | 486 | N/N | Alive | |
| 16 | M | 75 | 658 | N/N | Alive | |
| 17 | M | 78 | 862 | Y/Y | Alive | |
| 18 | M | 76 | 1,022 | Y/Y | Alive | |
| 19 | F | 82 | 1,056 | Y/Y | Alive | |
| 20 | M | 76 | 1,279 | Y/Y | Alive | |
| 21 | M | 79 | 1,442 | Y/Y | Alive | |

**Notes.**

HR, Hazard ratio; DM, Diabetes mellitus; HT, Hypertension; COPD, Chronic obstructive pulmonary disease.

[a]Survival time: from day of diagnosis to death or last follow-up dates.

however, 11 patients (11/16, 68.8%) completed the therapy. The surviving patients all completed the planned oncological treatments (either surgery alone, or surgery and adjuvant treatments).

## DISCUSSION

Our study showed that advanced T classification (T3–4), positive nodal metastasis (N1–3) and poorly differentiated tumor predicted poor survival for all patients, which were compatible with previous studies (*Aksu et al., 2006*; *Goto et al., 2005*; *Liao et al., 2008*). The male patients showed significantly poor survival than the female patients for all patients, but showed no significant difference after dividing all patients into early and advanced stages. This may be due to the relatively smaller sample size of female (1,504 men *vs* 208 women) in our study cohort. Previously published literature on the outcomes of the surgical treatment of the tongue cancer in different age groups has been controversial. (*Sarini et al.,*

*2001*) reported the treatment outcomes of older patients (≥75 years) with head and neck squamous cell carcinoma did not differ significantly from younger patients' outcomes. *Davidson, Root & Trock (2001)* reported a large series ($n = 749$) of the tongue cancer patients enrolled in the Surveillance, Epidemiology, and End Results (SEER) database concluded that an increasing age predicted the worse disease-specific survival. *Chang et al. (2013)* reported old patients (>65 years) with oral cavity cancer had lower survival rate than young patients (<45 years). However, no details of cancer staging were reported in these study (*Chang et al., 2013*; *Davidson, Root & Trock, 2001*). *Jones et al. (1998)* and *Clayman et al. (1998)* also reported that older patients with head and neck cancer had worse survival that younger patients. Nonetheless, oral cavity cancer patients comprised less than 25% to 60% of their patients.

In our study, we included 1,712 homogeneous tongue cancer patients (1,476 younger, 178 young old and 58 older old patients) with clear pathological staging after radical surgery, and compared their overall survival rate with younger patients all treated under the standard guidelines. Our study clearly showed that elderly patients are likely to face the worst survival rate amongst the tongue cancer patients after having been treated by radical surgery. After adjusting for other variables, young old and older old patients were more likely to die than younger patients. No significant difference in adjusted HR of death was found for early-stage patients (stage I–II) amongst the younger, young old, or older old patients which implied that age should not deny older people to receive optimal treatment. However, for advanced-stage disease (stage III–IV), the older old patients showed significantly worse survival than the other two groups after adjusting for other variables.

*Italiano et al. (2008)* reported on 316 head and neck squamous cell carcinoma patients aged >80 years receiving radiotherapy ($N = 180$; 57.0%), surgery ($N = 97$; 30.7%) and no treatment ($N = 39$; 12.3%). They reported that outcomes of patients with stage I/II was similar to that of younger patients, but those with stage III/IV showed poor survival. These results are in agreement with our results. In our series, younger, young old, and older old patients received similar treatment modalities (Table 2, 81.0–86.7% patients underwent surgery alone without CT or RT) and had comparably optimal survival rate in early-stage tongue cancer. Our data represented the first evidence that old age ≥75 years should not be a reason to deny patients of early-staged tongue cancer to receive curative surgery.

For advanced-stage patients, older old patients had worst prognosis as compared with the other two age groups. Fourteen out of 21 older old, advanced-staged patients finally died and most of the mortalities occurred within 1 year after cancer diagnosis (12/14, 85.7%). The causes of deaths were mostly non-cancerogenic (9/14, 64.3%) including underlying diseases in combination with infection, pneumonia, poor nutrition status, and fall-related injury.

*Reid et al. (2001)* concluded that comorbidities also predict survival in the older people with head and neck cancer. Previous studies have also emphasized the importance of careful assessment of comorbidities, physical status, and patients' psychological profiling before operation (*Grenman et al., 2010*; *Kruse et al., 2010*). Many studies have indicated regular physical activity is essential for the elderly cancer patients to aid in the process of recovery, improve fitness and prevent falls (*Cho et al., 2015*; *Genden et al., 2005*; *Keogh et al., 2015*; *Lee et al., 2016*; *Pinto & Ciccolo, 2010*; *Rock et al., 2012*). Besides, our results

(Table 2) showed that the proportion of the patients who received postoperative CCRT was significantly low in the elderly patients. Adjuvant RT or CT after surgery was indicated for eighteen of 21 older old, advanced-staged patients, and was received by 11 of those patients (61%). The reason for not receiving adjuvant therapy were advanced age ($n = 4$), comorbidities ($n = 2$), and early death ($n = 1$). Thus, suboptimal treatments might increase the risk of cancer recurrence and disease metastasis in cases with advanced disease.

The following measures possibly could improve the survival rate of the elderly patients with tongue cancer: (1) thoroughly evaluating patients pre-operationally and controlling the underlying disease (2) using geriatric assessment tools to predict mortality and assist treatment decision-making process (*Extermann & Hurria, 2007*; *Italiano et al., 2008*); (3) screening the cancer intensively to diagnose cancer as early as possible (*Reid, 2013*); (4) ensuring that patients receive post-operational rehabilitation for cancer-related deconditioning as soon as possible (*Saotome, Klein & Faux, 2015*); (5) increasing nutrition supplementation and preventing choking and aspiration pneumonia (*Farhangfar et al., 2014*); and (6) Modification of environmental hazards and performing physical activities to prevent falls which is common in older cancer patients (*Cho et al., 2015*; *Keogh et al., 2015*; *Lee et al., 2016*; *Rock et al., 2012*; *Sattar et al., 2016*; *Ungar & Rafanelli, 2015*).

Future research by incorporating these factors or measures should be considered in order to improve survivals in those patients.

## CONCLUSION

Our study showed that advanced T classification (T3–4), positive nodal metastasis (N1–3) and poorly differentiated tumor predicted poor survival for all patients. For early-stage patients (stage I–II), the overall survival rate among the younger age, young old, and older old patients were not significantly different. However, for advanced-stage patients (stage III–IV), the older old patients ($\geq 75$) had significantly worse survival than the other two patient groups. Based on the present study, we suggest that age should not deny early stage patients to receive optimal oncological treatment. However, older old patients ($\geq 75$) with advanced cancer should be comprehensively assessed by geriatric tools before surgical treatment combined with intensive postoperative care to improve survival.

## ACKNOWLEDGEMENTS

The authors would like to thank Center of Excellence for Chang Gung Research Datalink (CORPG6D0161-2, CORPG6D0251-2) for the comments and assistance in data analysis.

### Funding

The authors received no funding for this work.

### Competing Interests

The authors declare there are no competing interests.

## Author Contributions

- Ming-Shao Tsai conceived and designed the experiments, performed the experiments, wrote the paper.
- Chia-Hsuan Lai conceived and designed the experiments, performed the experiments.
- Chuan-Pin Lee analyzed the data, prepared figures and/or tables.
- Yao-Hsu Yang analyzed the data.
- Pau-Chung Chen, Re-Ming A. Yeh and Wen-Cheng Chen reviewed drafts of the paper.
- Chung-Jan Kang, Geng-He Chang, Yao-Te Tsai, Chih-Yen Chien and Ku-Hao Fang contributed reagents/materials/analysis tools.
- Chang-Hsien Lu wrote the paper, reviewed drafts of the paper.
- Chi-Kuang Young contributed reagents/materials/analysis tools, wrote the paper.
- Chin-Jui Liu prepared figures and/or tables.

## Human Ethics

The following information was supplied relating to ethical approvals (i.e., approving body and any reference numbers):

The ethics review board of our institution approved the study (CGMH-IRB No. 104-4642B).

## Data Availability

Mortality in tongue cancer patients treated by curative surgery: https://dataverse.harvard.edu/dataset.xhtml?persistentId=doi:10.7910/DVN/U0RZHZ.

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
