# Peer review of "Mortality in tongue cancer patients treated by curative surgery: a retrospective cohort study from CGRD"

_PeerJ, doi:10.7717/peerj.2794_

## Round 0.1 · original submission · Major Revisions

· Academic Editor

Major Revisions

The reviewers have highlighted a number of major areas in which this manuscript needs to be improved before it can be more strongly considered for publication in PeerJ. I would therefore suggest you look carefully at their comments before resubmitting the manuscript.

Reviewer 1 ·

Basic reporting

1. The research paper is well written, however, changes must be made to improve the problem of the study and the objectives of the research. Also, the three groups studied should be standardized across the text. In reporting results, I would like to see the confidence interval instead of P-value since the 95%CI would give me the idea how was the data.
2. The background is well written, but I feel like it needs some improvements. Having said that I recommend focusing on the problem and what you want to show with your research. I understood that your priority was comparing the three age groups. One example … to compare the outcomes of surgical treatment of the tongue cancer patients in three different age groups… What are the outcomes? I recommend you to report each single outcome that you will present, just saying an outcome is vague.
3. The text should be improved for clarity
4. Figures and table are fine. I recommend you to explain what is the P-value reported for. For example, you reported some significant P value in some tables, but I don´t know what is different from what and where is the difference.
5. Raw data is supplied.

Experimental design

1- The research is in the scope of the journal
2- As I mentioned above is not clear the research question. I don´t see the gap where the research fills it. Also, most of the references are ten years old, there is a recent study that could be cited about the topic that I found on Pubmed.
3- The investigation was well conducted.
4- Methods are well explained, but need to clearly define all outcomes, exposures, predictors, potential confounders, and effect modifiers.
5- The study needs to explain how the study size was arrived at this number.
6- What were the efforts to avoid bias? Authors should explain that.

Validity of the findings

1- The study is interesting, but need to improve reporting and clarity.
2- The conclusion of the study supports the findings. What I still don´t understand is the division in these three subgroups.

Additional comments

The study needs major revision and I think it has potential to be published

Reviewer 2 ·

Basic reporting

See general comments below.

Experimental design

See general comments below.

Validity of the findings

Conclusions and implications of the study do not match the study results. See general comments section below.

Additional comments

The paper looks at mortality in tongue cancer patients treated with curative surgery, finding that patients with advanced cancer, and older patients fared worse than their younger counterparts. The paper is written comprehensively, and is well presented.

MAJOR COMMENTS
1. A major issue of the paper is that it does not provide too much of an insight – merely that older patients, or those patients with advanced cancer die more….
2. The tone of the Discussion section is different from the rest of the paper – it is not humble in the slightest, and you have implied that your study is superior to others. It is suggested a more modest tone be set in the Discussion section.
o Line 240: “we believe our study is more convincing than previous reports” (please remove/rephrase this sentence)
3. What are the implications of your study for clinicians? This is not clear from your study. ie what does your study add to the literature? How will the results of your study be used by oncologist’s in their decision making? At the moment, the “big picture” of how your study impacts on clinical practice or guidelines is not in your study.
4. The last paragraph in your Discussion (lines 276-281) section talks about measures that would help the survival rate of older patients with tongue cancer. However, these suggestions are not based on the results of your study – rather they are author opinions. Moreover, these suggestions are not referenced.
It is suggested for this paragraph that you discuss what additional research is needed to reduce mortality rates in older cancer patients. “eg future research should consider the impacts of nutritional supplementation methods to prevent choking….”.
It would be good if you also expanded on screening for patients that may have their treatment stopped mid-way (eg I think this is what you are trying to say, based on lines 273-274 but it is not clear, nor referenced).
5. 15 authors is a lot for this type of study

MINOR COMMENTS
1. Results section – the information in your tables is duplication in the text of your Results section. This makes your Results section very wordy and difficult to read. It is advised that you just refer to what is higher/lower in your Results section, rather than repeating values from the Tables.
2. The word “subjects” is considered discriminatory. Please replace with the word “patients” or “participants”. Similarly, the word “elderly” is considered to be derogatory by the UN. Therefore, please replace the word “elderly” with a more appropriate term, such as “older people”, “older persons”, or “older adults”
3. There are a few minor grammatical errors, with “the” and “a” sometimes being mixed up/omitted. Same goes for “and” and “or”.
a. Line 190 – replace “or” with “and”
b. Line 138 – “regular postoperative” is correct terminology here
c. Line 150 – “them”? What is “them”? The Kaplan-Meier curves? The follow-up period? The age groups? Please be specific here.
d. Line 68. “The tongue”, rather than “Tongue”
e. Line 88 – Replace “the” with “a”
f. Please use units were applicable – eg lines 98 – 100 (“years”) – see point below
g. Lines 225 – 226: “…(Sarni et al. 2001) compared the treatment outcomes of 273…” is correct grammar here

4. Lines 98-100. Please rephrase this sentence so that it makes sense. Also place in past tense. “Here, we compared treatment results of tongue cancer patients, stratifying by age groups: <65 years (younger population), 65-<75 years (young old), and ≥ 75 years (older old and oldest old)”
5. Line 88 – what is worse than death? Please rephrase this sentence so it states what you want it to mean.
6. Lines 111 – 115: how many patients did you exclude for the various reasons?
7. Line 125 – this sentence needs a reference, or the guidelines listed in your Appendix section
8. Your discussion is reading like a bibliography. Eg Lines 228 – 233 need rephrasing – compare and contrast results of other papers with your paper directly, rather than listing what other papers have done.
9. Lines 256 – 259 are confusing. Also remove the word “enjoy” – you did not ask for patient satisfaction or quality of life in your study.
10. Lines 273 – 274. This sentence needs to be rephrased. Presently, it reads as though it is the older patient’s fault that they died during surgery. Also what does “poor overall condition” mean? Are you referring to frailty? The patient’s sociodemographics? Or advanced age? How was “poor overall condition” assessed? If this was “eyeballed” rather than measured objectively, then this sentence needs to be removed entirely, or at least referenced.
11. Line 281 – please remove the word “great” as it implies that current nursing systems are not great.
12. Line 281 – falls: it would be advantageous if you discussed this in more detail (with a reference or two)
13. Line 392 – reference: space needed in the author name here. Also, reference on line 385 has an error in the author. Same goes for reference on line 390. I think you may need to redo your references in your reference database for non-journal articles.

---

## Round 0.2 · Minor Revisions

· Academic Editor

Minor Revisions

Thank you for your diligence in this review process. You are to be congratulated as you have made almost all of the requested changes to the manuscript from the two reviewers. A small number of minor issues remain prior to the paper being accepted for publication, with these being:

Line 287: Remove the Davidson reference from the citation.
Line 281 – 282: change “mortalities were occurred” to “mortalities occurred”.
References: there still appear to be some issues regarding the consistency of references that are cited in the text, with some having a space and others between the citation and the sentence. Further, there seems to be some inconsistencies in the manner in which the Journal names are presented in the reference list. Please check for consistency in the use of the journals full name/abbreviation And How the journal name is capitalized.

Reviewer 1 ·

Basic reporting

- All changes I have asked were done.

Experimental design

- All changes I have asked were done.

Validity of the findings

- All changes I have asked were done.

Additional comments

- All changes I have asked were done.

---

## Round 0.3 · accepted · Accept

· Academic Editor

Accept

We thank you for making the necessary changes to the manuscript.